# Model design choices impact biological insight: Unpacking the broad landscape of spatial-temporal model development decisions

Jessica S. Yu [1,2], Neda Bagheri [1,2,3]*

**1** Chemical and Biological Engineering, Northwestern University, Evanston, Illinois, United States of America, **2** Biology, University of Washington, Seattle, Washington, United States of America, **3** Chemical Engineering, University of Washington, Seattle, Washington, United States of America

* nbagheri@uw.edu

**Data Availability Statement:** All raw simulation data and results are available on Mendeley Data: System representation (https://doi.org/10.17632/bh5w3tzsjc), Cell-to-cell variability (https://doi.org/

## Abstract

Computational models enable scientists to understand observed dynamics, uncover rules underlying behaviors, predict experimental outcomes, and generate new hypotheses. There are countless modeling approaches that can be used to characterize biological systems, further multiplied when accounting for the variety of model design choices. Many studies focus on the impact of model parameters on model output and performance; fewer studies investigate the impact of model design choices on biological insight. Here we demonstrate why model design choices should be deliberate and intentional in context of the specific research system and question. In this study, we analyze agnostic and broadly applicable modeling choices at three levels—system, cell, and environment—within the same agent-based modeling framework to interrogate their impact on temporal, spatial, and single-cell emergent dynamics. We identify key considerations when making these modeling choices, including the (i) differences between qualitative vs. quantitative results driven by choices in system representation, (ii) impact of cell-to-cell variability choices on cell-level and temporal trends, and (iii) relationship between emergent outcomes and choices of nutrient dynamics in the environment. This generalizable investigation can help guide the choices made when developing biological models that aim to characterize spatial-temporal dynamics.

## Author summary

Scientists often rely on computational models to identify the basic building blocks, and uncover interactions, underlying complex biological systems. These models have proven invaluable at enabling researchers to computationally test, refine, and generate new hypotheses that can then be experimentally validated. The impact of computational models is constrained by what we do not know about the biology and the limitations of our computer software and hardware. Scientists have to make choices about what to include or leave out in their models because of these limitations.

10.17632/dvrs8zsngb), Nutrient dynamics (https://doi.org/10.17632/96jtpmnrgt). All source code for the model is available on GitHub at https://github.com/bagherilab/ARCADE. The scripts used to perform analyses and generate figures reported in this paper are available on GitHub at https://github.com/bagherilab/arcade_modeling_choices.

**Funding:** This work was supported by the National Science Foundation Graduate Research Fellowship Program award DGE-1842165 (JSY), the National Science Foundation CAREER award CBET-1653315 (NB), and the Washington Research Foundation (NB). The funders had no role in study design, data collection and analysis, decision to publish, or preparation of the manuscript.

**Competing interests:** The authors have declared that no competing interests exist.

In our study, we investigate the impact of select modeling choices on the outcomes of an agent-based model characterizing cell population dynamics. We focus this study on three types of modeling choices that are not often rigorously evaluated despite their prevalence across many different types of models: (1) system representation, (2) cell/agent variability, and (3) environment dynamics. We report how each of these modeling choices affects simulated emergent outcomes, specifically growth rate, symmetry, and cell cycle length. Our research emphasizes how crucial it is to carefully consider and communicate the impact of these choices when developing computational models.

## Introduction

Computational models are a critical framework for interrogation and discovery of biological phenomena. Computational models enable hypothesis generation, hypothesis testing, analysis of state and parameter spaces, and greater mechanistic understanding of biological systems [1, 2]. Given the complexity of biological systems, these models necessarily make widespread assumptions and abstractions to ensure feasibility. There exist fundamental modeling trade-offs between realism, precision, and generality; these trade-offs are governed by specific system contexts [3, 4]. For instance, a researcher may choose to include realistic mechanistic behavior for a specific cell type by developing a precise model that accurately captures a behavior of interest. However, analogous to overfit machine learning models that perform poorly on data outside its training set, a computational model developed and parameterized to represent a specific cell type within a specific environmental context may not be generalizable to other cell types or contexts. Alternatively, a researcher may choose to simplify mechanistic behavior into more abstract systems of equations that produce precise results, but assume ideal conditions (e.g. no cell-to-cell variability or homogeneous environmental conditions). These models can be generalized to other cell types or contexts, but might not characterize realistic phenomena. Finally, a researcher may choose to integrate abstract cell rules with more realistic (non-ideal) assumptions at the cost of gaining qualitative, rather than precise quantitative, insights. Balancing these trade-offs is essential for designing models that produce relevant outcomes with respect to imposed constraints.

In model development, decisions about these trade-offs are often made heuristically and reactively rather than deliberately by design. To elucidate the impact these design decisions have on model outcomes and inform future model design decisions, we investigate how common modeling choices influence resulting emergent behavior in context of an agent-based model (ABM). We define emergent behavior as a quantifiable property of any system that is not pre-defined and does not arise directly from parameters of constituent components.

ABMs are a bottom-up modeling framework in which autonomous cell agents follow a set of rules that guide their actions and interactions [5]. The ABM framework is extremely flexible for testing different modeling choices at the system, cell, and environment levels. We specifically investigate how decisions regarding system representation, cell-to-cell variability, and nutrient dynamics impact simulation outcomes using the ABM framework ARCADE as a testbed [6]. ARCADE simulates the growth of cancer cells in a simple tumor microenvironment, where cells are represented as autonomous agents that follow rules guiding transitions between different cell states: proliferative, migratory, quiescent, apoptotic, necrotic, and senescent. Each cell comprises a metabolism module and a signaling module, which couple the cell to their local environment through nutrient consumption and EGFR signaling.

ARCADE provides two environmental contexts for simulating cancer growth: colony and tissue. The *colony* context comprises only of a cancer cell population that reflects the hallmarks of cancer to represent tumor growth *in vitro*. The *tissue* context introduces a background healthy cell population to represent tumor growth *in vivo*. In both contexts, cancer cells are introduced to the center of the simulation environment at the start of the simulation. Prior work highlighted differences in emergent behavior between colony and tissue contexts [6, 7]. In this study, we perform simulations under both contexts to evaluate whether the influence of modeling choices on emergent behavior is consistent between the two.

In order to compare simulations and analyze differences as a function of modeling choices, we define three general summary metrics that quantify emergent behaviors: growth rate, symmetry, and cell cycle length. Growth rate is a temporal emergent property; cell growth is governed by local nutrient uptake rather than a global growth rate parameter. Symmetry is a spatial emergent property; cell migration follows rules based on local neighbor density and volume constraints. Cell cycle length is a single-cell-level emergent property; cell division is a function of growth and migration rules and is not explicitly set as a model parameter.

Results on how choices of system representation, cell-to-cell variability, and nutrient dynamics impact simulation outcomes aim to guide researchers on how to navigate modeling trade-offs and proactively design models to produce insights that are relevant to both the research question and the system of interest.

## Results

### System representation: Model geometry and dimensionality

There are many ways to represent the system when building a computational model of a biological system. We consider two system representation choices—geometry and dimension—that are commonly made across many modeling frameworks (Fig 1A). *Geometry* refers to the coordinate system used in a model. We specifically compare rectangular and hexagonal geometries, but note that alternative geometry choices exist. Many epithelial tissue cells pack in a

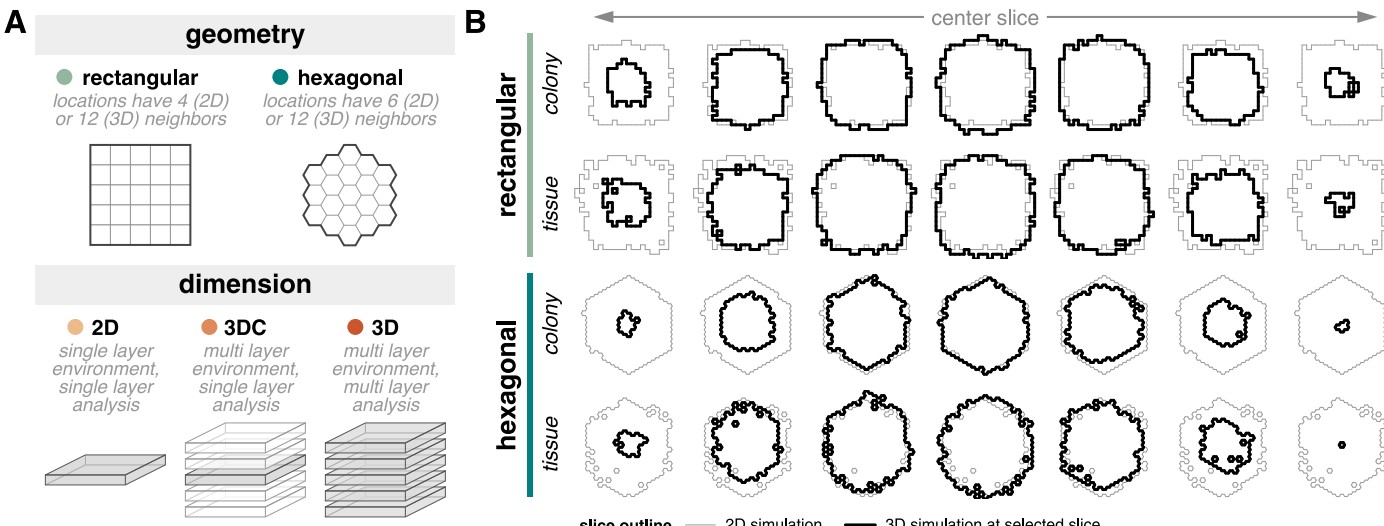

**Fig 1. System representation simulations.** (**A**) Diagram of geometry and dimension model choices for system representation simulations. Geometry describes the choice between rectangular or hexagonal coordinates. Dimension describes the choice between 2D, 3D, and "center slice" of 3D (denoted 3DC) simulations. (**B**) Outline of colonies for 2D (thin, gray lines) and 3D (thick, black lines) simulations for different geometries and contexts. Only select slices of the 3D simulation are shown.

hexagonal or near-hexagonal geometry [8, 9] while computational models often utilize a rectangular geometry for reasons including simplicity of model formulation, convenient mathematical properties, and reduced computational resource usage. *Dimension* refers to the dimensionality of a model. Increasingly, experimental studies have highlighted differences between cells grown in 2D and 3D culture and the need for 3D models to help uncover rules underlying differences between *in vitro* and *in vivo* behavior [10–12]. We also consider the "center slice" of 3D models (denoted 3DC) as an intermediate dimension choice [13].

**System representation choices balance accurate representation with computational cost.** We simulated all combinations of geometry and dimension for both colony and tissue contexts (S1(A) Table). All simulations utilize a constant nutrient source environment to reduce heterogeneity introduced by environmental variation. All simulations are initialized with variable cell ages and volumes; initial volumes are drawn from a normal distribution and initial ages are drawn from a uniform distribution.

Each system representation presents distinct benefits and drawbacks. While one option might be computationally simpler (rectangular; 2D), another might be more biologically relevant and also more complex (hexagonal; 3D). Using CPU time as a proxy for complexity confirms that the former choices represent the less complex options (S2(A) Table). Understanding which and how these model design choices impact emergent behaviors of interest should guide the deliberate development and design of computational models.

Broadly, there is more variation in tumor shape for cells grown in tissue context compared to those in colony context, but the resulting tumors are generally similar in size across system representation choices (S1(A) Fig) and comprise the expected quiescent core and proliferative rim (S1(B) Fig). The colony shape in 2D simulations are most consistent with the colony shape in the center slice of 3D simulations (Fig 1B). Overall, system representation model choices do not have a major impact on the simulated tumor population. However, more subtle differences in emergent simulation outcomes do exist as a function of system representation choices.

**System representation choices drive quantitative changes in emergent behavior.** For different choices of geometry and dimension, the trends in emergent behavior metrics are qualitatively similar over time (S2 Fig). However, quantitative differences do arise as a function of system representation choice, simulation context, and specific emergent behaviors of interest. An experimental study provides an illustration of this observation; researchers identified a difference in infection sensitivity between tissue culture and mouse models, but also revealed a statistically significant correlation between the two contexts [14].

For dimension choices, growth rate is similar between 2D, 3DC, and 3D, with slightly lower growth rate in 2D for rectangular simulations (S2(A) Fig). Symmetry metrics are consistent between 2D and 3DC; 3D tends to have lower symmetry. In contrast, cell cycle length is similar between 3DC and 3D; 2D tends to have longer cell cycle durations. These relative relationships among dimension choices hold between tissue and colony contexts.

For geometry choices, growth rate is higher and symmetry is lower in hexagonal coordinates than in rectangular coordinates (S2(B) Fig). We note that the symmetry metric is, by definition, a function of the geometry and therefore not directly comparable between these geometry choices. Early cell cycle length is longer in hexagonal simulations for colony context simulations, but longer in rectangular simulations for tissue context simulations. These context-dependent differences in cycle length suggest that local competition for space and nutrient resources play an important role during initial tumor growth.

To better assess the statistical significance of observed differences, we use ANOVA to evaluate and compare metrics for emergent behavior (Fig 2, S3 and S4 Tables).

For growth rate, there exists a significant interaction effect between choice of geometry and dimension for both contexts (Fig 2A) suggesting that the choice of geometry and choice of

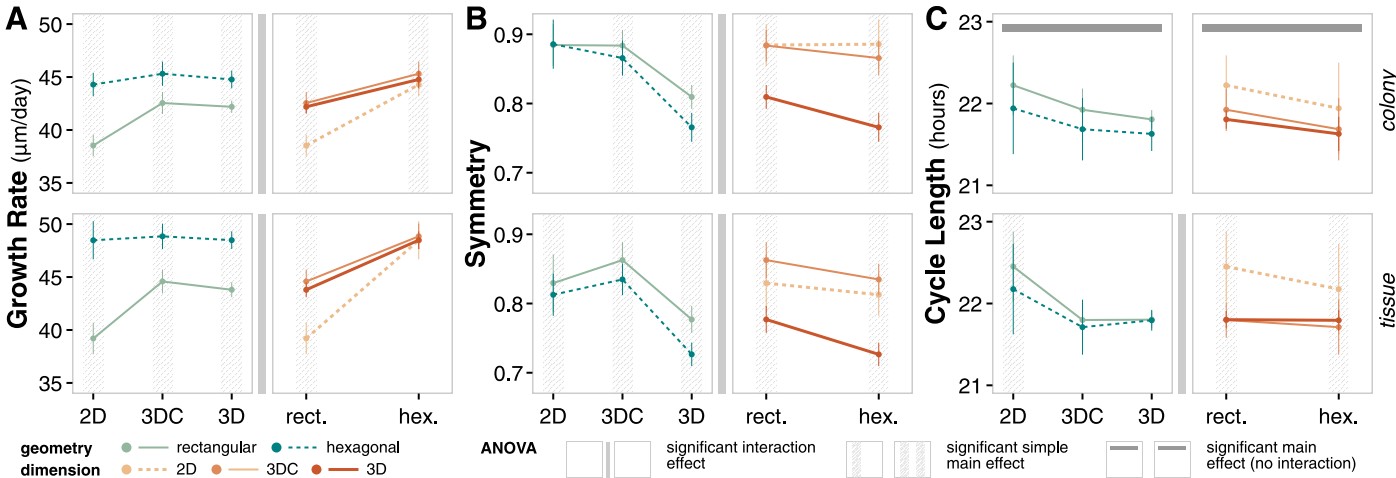

**Fig 2. System representation emergent behavior.** (**A**) Interval plots and ANOVA summary for growth rate at t = 14 days. Data points reflect the mean values of the metric at the final time point of the simulation for the given factor level; error bars show standard deviation of the metric. Significant interaction effects identify conditions where the effect of the choice of geometry on the metric is not independent from the choice of dimension. Simple main effects identify significant differences in metric between choices of geometry for a given choice in dimension, and vice versa. Main effects identify significant differences in metric between choices of geometry independent of choice in dimension, and vice versa. (**B**) Interval plots and ANOVA summary for symmetry at t = 14 days. Analogous to panel A. (**C**) Interval plots and ANOVA summary for cycle length at t = 14 days. Analogous to panel A.

dimension do not have independent effects on growth rate. In rectangular coordinates, there is clear distinction between 2D and 3DC/3D growth rate while the differences are less apparent in hexagonal coordinates. Therefore, 2D and 3D simulations in hexagonal geometry are more similar than 2D and 3D simulations in rectangular geometry.

For symmetry, there exists a significant interaction effect between geometry and dimension choices for both contexts (Fig 2B). Symmetry outcomes are significantly different among dimensions for both choices of geometry except for cells in a rectangular geometry and colony context. In this unique case, the symmetry of colonies in a 2D simulation are nearly the same as the center slice of a colony in 3D. Computationally, this similarity in symmetry suggests that a 2D simulation is representative of the spatial emergence of the center slice of a 3D simulation, enabling simplified modeling. However, the significant differences observed when considering the 3D colony as a whole or introducing complexity through tissue contexts or hexagonal geometry highlights that simple 2D, rectangular simulations in contexts that lack nutrient or spatial competition may obscure important spatial behavior.

For average cell cycle length, there is no significant interaction effect between choice of geometry and choice of dimension in the colony context, but there is a significant interaction effect in the tissue context (Fig 2C). This result suggests that system representation choices may independently impact emergent behavior under simpler contexts, but that the impact of these choices can become coupled in more complex contexts. Cycle lengths are significantly different between geometries for all choices of dimension except 3D/3DC in tissue context and between 2D and 3D/3DC for either choice of geometry. The difference between cycle lengths in 2D and 3D/3DC in particular highlight the impact of dimension choice on local nutrient conditions that drive emergent single cell behavior.

Overall, both geometry and dimension are important system representation choices in the development of a cell population model. While choices in geometry and dimension produce qualitatively similar trends in this system; there are quantitative differences that are important to consider when developing models to address specific questions. In some cases, a simpler

system representation may be sufficient; in others, a more complex system representation may be necessary.

## Cell-to-cell variability: Initial cell age and volume distributions

**Cell-to-cell variability choices balance realistic initial conditions with unnecessary noise.** Computational simulations often make the simplifying assumption of homogeneous cell populations, but there is fundamental and unavoidable heterogeneity between cells. Here, we focus on inherent cell-to-cell variability within a population rather than cellular heterogeneity deriving from genotypic differences. Specifically, we interrogate the initial distributions of two model-agnostic features—cell volume and cell age—that are relevant for both equation-based and rule-based modeling frameworks (Fig 3A). *Cell volume* captures the initial distribution of cell volumes, which is relevant to, and a proxy for, cell cycle progression. We compare emergent metrics from simulations of cells that start at the same volume and simulations of cells with initial volumes drawn from a normal distribution. Cell cycle synchronization enables researchers to study cell cycle regulation and control, while asynchronous populations reflect unperturbed growth [15, 16]. *Cell age* captures the initial distribution of cell ages, which is relevant to division history, accumulation of mutations, and self-renewal. Similarly, we compare emergent metrics from simulations of cells that all start at age zero and simulations of cells with initial ages drawn from a uniform distribution. Research in aging has suggested a common origin to cancer, identifying analogous hallmarks of aging and phenotypic changes associated with cell age [17, 18]. Simulations initialized with uniform cell properties reflects an ideal, but arguably unrealistic, synchronized population, while simulations initialized with variable cell properties reflects more realistic conditions.

We simulated all combinations of volume and age variability for both colony and tissue contexts (S1(B) Table). All simulations utilize a constant source environment with a 2D, hexagonal representation.

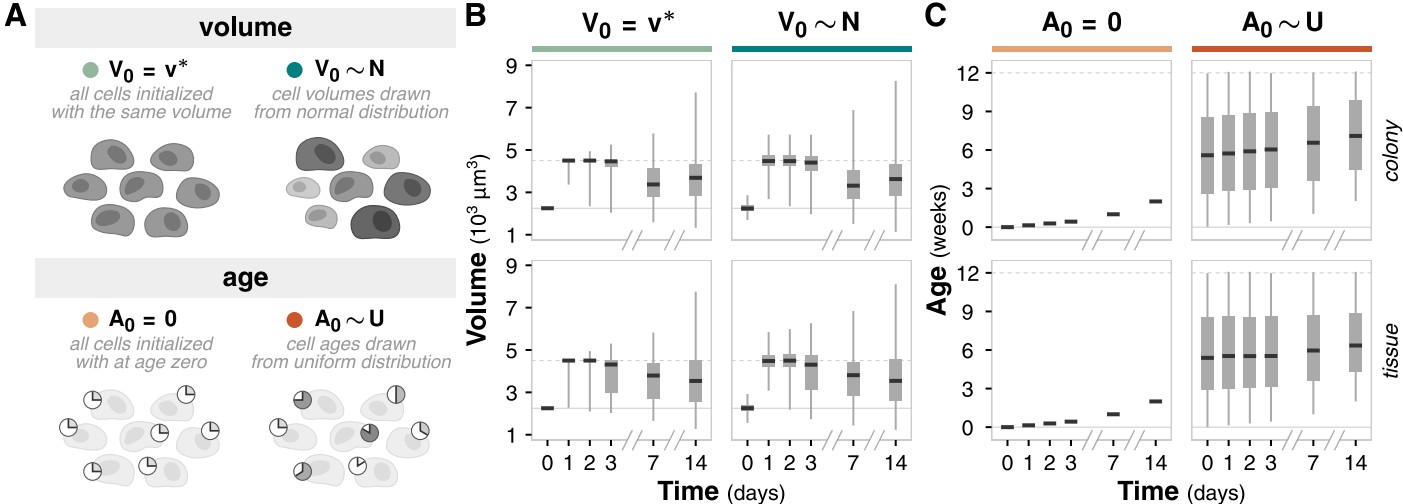

**Fig 3. Cell variability simulations.** (**A**) Diagram of volume and age model choices for cell-to-cell variability simulations. Volume describes the choice between initial cell volumes set to volume $v^* = 2250\ \mu m^3$ or drawn from the normal distribution $N(\mu = 2250\ \mu m^3, \sigma = 200\ \mu m^3)$. Age describes the choice between initial cell ages set to zero or drawn from the uniform distribution $U(0, 12\ weeks)$. (**B**) Box and whisker plot showing volume distribution over time. Black bar denotes median values; box denotes the lower and upper quartiles; whiskers denote the minimum and maximum volumes. Solid gray horizontal line indicates the average cell volume and dotted gray horizontal line indicates the average doubled cell volume. (**C**) Box and whisker plot showing age distribution over time. Analogous to panel B. Solid gray horizontal line indicates age 0 and dotted gray horizontal line indicates the maximum cell lifespan.

Choices in modeling cell age and volume variability highlight a fundamental question when building computational models: how much detail is necessary? Unlike system representation choices of dimension and geometry, these variability choices are decoupled from computational cost (S1(B) Table). The more "realistic" model choice (where initial cell volume and age are variable) is not necessarily better at addressing specific questions. Understanding how choices in cell-to-cell variability impact emergent behavior can help distinguish between cases where characterizing such variance is fundamental for addressing the question of interest and cases where the added variance merely dilutes observations.

Tumor shape is similar across all combinations of volume and age variability; more differences in shape are observed in the tissue context than the colony context (S3(A) Fig). Simulations with variance in initial age result in larger colony shape differences because cells initialized with higher ages apoptose earlier in the simulation. All combinations of initial cell-to-cell variability exhibit the expected quiescent core and proliferative rim (S3(B) Fig). For cell volume, there is no noticeable difference in the overall volume distribution after one week, although simulations initialized with variable volume exhibit slightly wider tails (Fig 3B). When a cell divides, the daughter cells do not inherit exactly half the volume of the mother, which introduces additional variation in the volume distribution over time. For cell age, simulations initialized without age variation produce a linear increase in age (from the imposed initial condition of age zero), while simulations initialized with age variation maintain the initial age distribution with a cutoff at the cell life span (Fig 3C).

**Cell-to-cell variability choices impact cell-level and temporal emergent behaviors.**
Between age variability choices, trends in growth rate, symmetry, and cycle length over time are quantitatively consistent (S4(A) Fig). Simulations in tissue context exhibit slightly higher growth rate, lower symmetry, and shorter cycle lengths, with differences becoming more apparent at later time points.

Between volume variability choices, trends in growth rate and symmetry are quantitatively consistent (S4(B) Fig). Cycle length is noticeably shorter for simulations initialized with variable volumes in colony contexts after about four days, possibly due to the faster division of cells initialized at smaller critical volume.

We use ANOVA to further assess and identify significant differences in emergent behavior metrics (Fig 4, S5 and S6 Tables).

For growth rate, symmetry, and cycle length, there is no significant interaction effect between the choice of initial volume variability and the choice of initial age variability for either context, which indicates that these two choices have independent impact on emergent behavior. In general, there also does not exist a significant main effect of initial volume variability on any emergent metric; the exception is cycle length in a colony context (as noted previously).

In contrast, the choice of initial age variability does have a significant main effect for all emergent metrics and contexts. Simulations initialized with variable cell age exhibit lower growth rates as more cells apoptose, lower symmetry as the apoptosing cells impact the shape of the colony, and shorter cell cycles as the apoptosing cells reduce the nutrient and spatial competition for proliferative cells.

Overall, both initial cell volume and age variability seem to have more of an impact on cell-level and temporal trends than population-level and spatial trends, and their impacts are independent in this system. Cell volume distributions become indistinguishable over time regardless of initialization, highlighting that the time at which measurements are taken is important: initial conditions strongly impact early spatial-temporal dynamics, but their effect wanes as the system reaches homeostasis at later time points. In contrast, cell age distributions are maintained over time, but how they are coupled to other behaviors in the model (such as probability

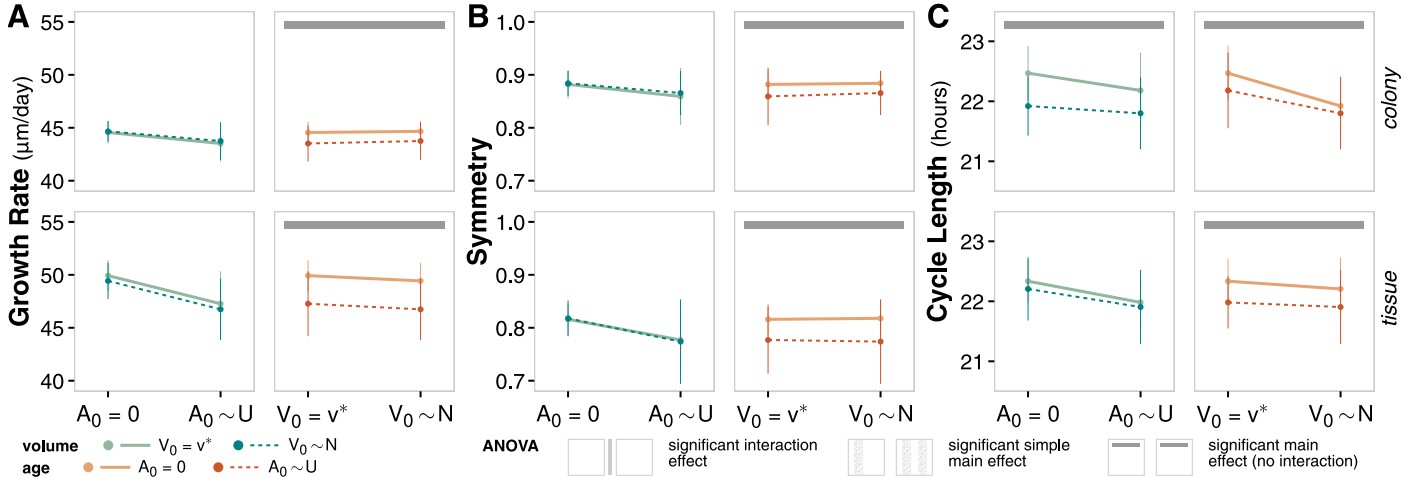

**Fig 4. Cell variability emergent behavior.** (**A**) Interval plots and ANOVA summary for growth rate at t = 14 days. Data points reflect the mean values of the metric at the final time point of the simulation for the given factor level; error bars show standard deviation of the metric. (**B**) Interval plots and ANOVA summary for symmetry at t = 14 days. Analogous to panel A. (**C**) Interval plots and ANOVA summary for cycle length at t = 14 days. Analogous to panel A.

of apoptosis, senescence, or accumulation of mutations) is important to consider depending on the context of the biological question. Questions involving cell-level or early temporal dynamics may need to account for more realistic, variable initial cell properties, while questions involving population-level or spatial dynamics may find the additional variability a source of unnecessary noise.

## Nutrient dynamics: Temporal profiles and concentrations

**Nutrient dynamics choices balance relevant environmental conditions with disproportionate model complexity.** The microenvironment is a critical and complex component of biological systems that presents multilateral regulation across scales. We have previously explored the impact of dynamic vascular architecture and hemodynamics on tumor growth and found that even simple changes in the nutrient environment can affect emergent behavior [7]. Here, we consider the impact of glucose temporal profiles and concentration levels as a proxy for microenvironmental nutrient dynamics (Fig 5A). *Profile* refers to the temporal shape of the glucose concentration. Often, glucose is represented as a constant value, but external perturbations such as media changes (represented by the "pulse" profile) and daily diet fluctuations (represented by the "cyclic" profile) drive temporal dynamics. *Level* refers to the average glucose concentration level; we vary the level from low to basal to high. These levels can vary physiologically due to disease, or experimentally as part of specific protocols.

We simulated all combinations of glucose profiles and levels for both colony and tissue contexts (S1(C) Table). All simulations utilize a 2D, hexagonal representation, with variable initial cell ages and volumes.

Nutrient dynamics are much more complex than the simplifications tested here. Physiological systems have complex vascular architectures and hemodynamics that affect nutrient delivery. Fluctuations in glucose due to daily food consumption are more heterogeneous than a simple cyclic function, and physiological conditions (such as diabetes) comprise more nuanced and complex mechanisms than simple changes in blood glucose levels. Fortunately, we do not need a comprehensive mechanistic model to interrogate nutrient dynamics impact emergent behavior. Complex vascular models can increase computational cost; the nutrient

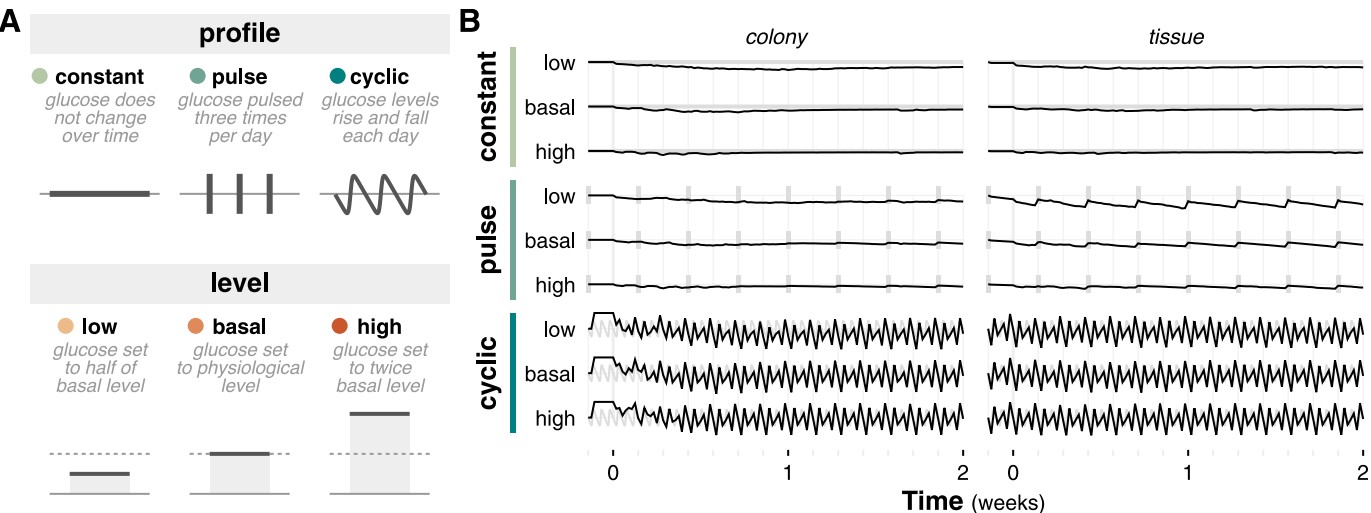

**Fig 5. Nutrient dynamics simulations. (A)** Diagram of temporal profile and level model choices for nutrient dynamics simulations. Nutrient profile describes the choice between constant, pulse, and cyclic dynamics. Nutrient level describes the choice between low, basal, and high glucose concentrations. (**B**) Sparklines of glucose concentration at the center of the simulation over time. Thick gray lines denote the nutrient profile at each time point.

dynamics choices described here do not have a noticeable impact on computational cost (S2 (C) Table). Understanding how nutrient dynamics affect emergent behavior can inform if and when these dynamics are relevant for a specific system.

Tumor size is strongly impacted by glucose level: higher glucose levels lead to larger tumors (S5(A) Fig). Tumors grown in tissue context exhibit more variation in shape. Simulations with low, cyclic glucose in tissue contexts were unable to successfully form tumors due to insufficient nutrients. Aside from this case, the subset of simulation conditions that give rise to tumor formation exhibit the expected quiescent core and proliferative rim (S5(B) Fig). The impact of temporal profile is clearly observed in nutrient concentrations over time (Fig 5B). Pulse and cyclic profiles result in more dynamic concentrations, which increase as the glucose level decreases and glucose becomes more limited. The cyclic profile in particular is very "noisy" and the large variation in glucose concentration over time indicates that measurement frequency must match the expected variation frequency in order to accurately capture nutrient dynamics. We note that the impact of glucose level is not clearly observed in nutrient concentrations over time due to the faster time scale of diffusion relative to the simulation time step (Fig 5B).

**Nutrient dynamics choices significantly impact emergent behaviors.** Trends in emergent behavior are highly dependent on glucose levels (S6(A) Fig). Overall, higher glucose levels correspond to faster growth rates and shorter cycle lengths. With symmetry, basal and high glucose levels show similar qualitative and quantitative behavior while low glucose levels show notably altered trends in combination with pulse and cyclic profiles.

Across nutrient profiles, trends in emergent behavior are highly independent, except with low glucose levels (S6(B) Fig). Differences in emergent behaviors between different profiles at low glucose are further magnified in the tissue context, where cells experience increased competition for nutrients and space.

We use ANOVA to analyze the significance of nutrient dynamics choices on emergent behavior metrics (Fig 6, S7 and S8 Tables).

For growth rate, there exists a significant interaction effect between nutrient profile and level for both contexts, although the effect size is more apparent in the tissue context (Fig 6A).

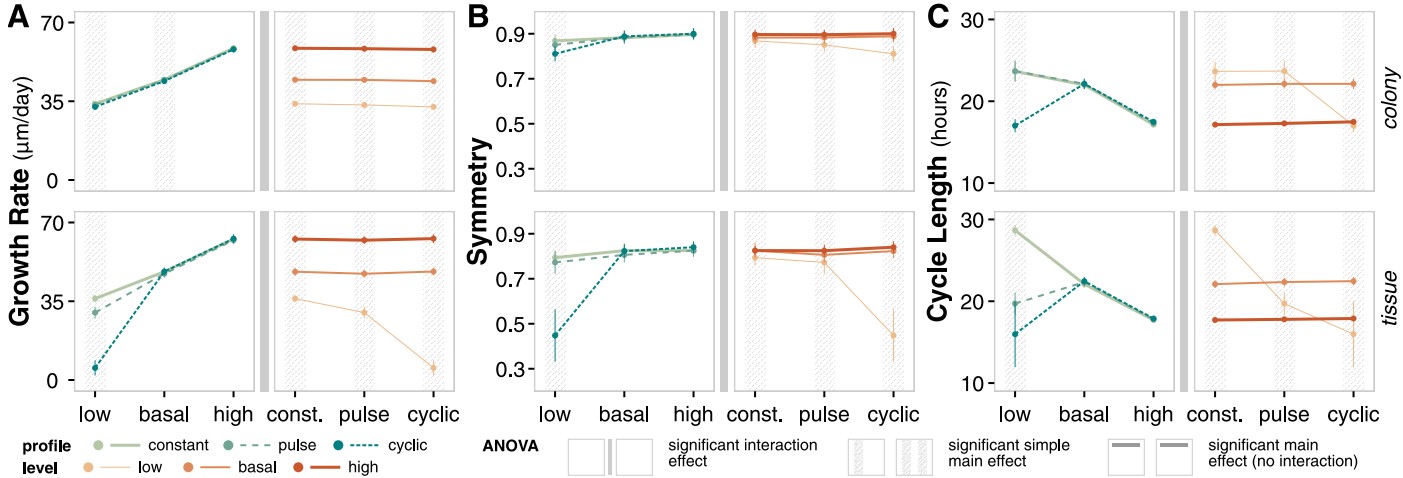

**Fig 6. Nutrient dynamics emergent behavior.** (**A**) Interval plots and ANOVA summary for growth rate at t = 14 days. Data points reflect the mean values of the metric at the final time point of the simulation for the given factor level; error bars show standard deviation of the metric. (**B**) Interval plots and ANOVA summary for symmetry at t = 14 days. Analogous to panel A. (**C**) Interval plots and ANOVA summary for cycle length at t = 14 days. Analogous to panel A.

There are clear differences in growth rate at different glucose levels; differences in growth rate as a result of nutrient profile are more apparent in the low glucose and tissue context. The impact of nutrient dynamics choices are more apparent in systems where nutrients are a limiting factor (e.g. tissue context).

For symmetry, there also exists significant interaction effects between nutrient dynamics choices (Fig 6B). In particular, the cyclic profile with low glucose levels results in tumors with significantly lower symmetry compared to other profiles at low glucose levels and compared to the cyclic profile at basal and high glucose levels.

The choice of nutrient profile and level have the greatest effect on emergent cycle lengths, combining the clear differences as a function of level observed with growth rate with the distinct behavior of the low/cyclic case observed with symmetry (Fig 6C). Context also plays an important role under specific combinations of profile and level choices; simulations with constant nutrient profile at low levels have much longer cycle lengths in tissue compared to the same simulations in colony context. In contrast, profile choice at basal and high levels have little impact on cycle length regardless of context.

Overall, choices on nutrient dynamics have a significant impact on emergent outcomes in this system, particularly when the choices result in nutrient-limited conditions. The choice of a static, constant source of nutrients at a basal level (or omitting environmental considerations entirely) excludes a significant source of temporal and spatial variation in biological systems as cells do not exist in isolation. That said, a complex model that includes detailed vasculature is not always necessary. Deliberate choices on nutrient dynamics and analysis of the sensitivity of models to environmental conditions can inform when environmental complexity is necessary to accurately interrogate the system of interest.

## Discussion

Computational models are a ubiquitous tool for the study of biological systems. It is critical to make deliberate and informed choices when developing these models to effectively represent the system and address the question of interest. Rigorous model development practices should include (and report) an assessment of the impact of modeling choices on the simulated

behavior of interest. Similar to traditional model analyses (e.g., robustness analysis and sensitivity analysis [19]), these assessments enable the modeler to better understand and refine both the structure and utility of their model for its designed purpose. In this study, we identified and interrogated how select modeling choices—at the system, cell and environment levels—impact observed temporal, spatial, and single-cell-level emergent behaviors in an ABM of cancerous cell growth. It is important to note that the assessment of modeling choices is not a form of model validation and does not require experimental data. Our modeling choices are agnostic to the specific modeling framework, and we are able to identify broadly applicable takeaways for future models developed in this, or similar, contexts. However, given that all models are both question and context dependent, the conclusions derived from this study are not necessarily generalizable across all computational models; this reality further motivates the need to interrogate modeling choices as a necessary step in the formal model development process.

First, we consider system representation choices (model geometry and dimension) and demonstrate their quantitative, and generally not qualitative, impact on observed behaviors. Second, we consider cell-to-cell variability choices (initial distributions of cell age and cell volume) and demonstrate how initial variability has more of an impact on cell-level and temporal trends than population-level and spatial trends. Finally, we consider nutrient dynamics (temporal profile and level of glucose in the environment) and highlight the key role of nutrients and the tumor microenvironment on emergent outcomes [7].

The ABM framework used to explore these modeling choices is readily extensible for further interrogation. The framework can be used to explore how duration of simulations, spatial and temporal resolution, or sampling frequency can impact model outcomes. Complexity can be added to explore additional modeling choices at the system, cell, and environment levels. For example, the current system representation choices comprise regular grids, but more complex geometries or continuous domains could be introduced and tested. Furthermore, additional emergent behavior metrics relevant to specific questions could be assessed. While ABM frameworks have become increasingly popular for characterizing biological systems, this evaluation can be applied to diverse model types to evaluate the impact of specific and relevant modeling choices.

Computational exploration of modeling choices can also be used to guide experimental design. For example, comparing simulations between 2D and 3D can identify key differences that translate to differences in experiments performed in 2D vs. 3D cultures. Differences in behavior due to initial cell-to-cell variability can identify cases in which additional synchronization or sorting of cells would be appropriate for specific experiments. Models can be designed to mimic an experimental setting, such as a tissue culture, and assess how media changes may affect cell dynamics.

Finally, we note that these simulations were not tuned to a specific system nor did we include highly detailed or complex mechanistic rules. The goal of this study is exploratory. We identified and tested important, but often not deliberate, choices made when developing computational models to identify which, when, and why certain choices may warrant deeper consideration during the model building process.

## Methods

### Model framework

We use our previously published agent-based modeling framework ARCADE [6, 7]. The framework contains three main packages: simulation (`sim`) to manage simulation inputs, outputs, and running the simulation itself, agents (`agent`) to represent physical cells, subcellular

behaviors, and other perturbations, and environment (`env`) to represent agent locations, nutrient diffusion, and environmental components such as vasculature.

**Model framework modifications.** Minor modifications were made to ARCADE to enable comparisons of different modeling choices (S7(A) Fig).

**Modifications for 3D simulations.** ARCADE is able to handle 3D simulations by stacking layers of 2D simulations (S7(B) Fig). Cell agents are able to move freely between layers and nutrient diffusion is calculated in 3D. An additional flag (`--view`) was added to ARCADE to enable visualization of the 3D simulations by averaging across the z dimension. Visualization is handled by the new classes `GrowthVisualization3D`, `AgentDrawer3D`, and `EnvDrawer3D`.

Note that while all source and pattern sites work for 3D simulations, the existing graph site representation of vasculature does not work for 3D simulations.

**Modifications for rectangular geometry.** ARCADE originally supported only hexagonal geometry. The equivalent rectangular geometry was added via a number of parallel classes: `RectLocation`, `RectAgentGrid`, `RectEnvLattice`, `RectDiffuser`, `RectPatternSites`, and `RectGraphSites`. In addition, the `GrowthSimulation` class was updated to initialize `Rectangular` simulations. All corresponding 2D and 3D visualization classes (`GrowthVisualization2D`, `GrowthVisualization3D`, `AgentDrawer2D`, `EnvDrawer2D`, `AuxDrawer2D`, `AgentDrawer3D`, and `EnvDrawer3D`) were updated to support visualization of rectangular geometry. A comparison of rectangular and hexagonal geometries is given in S7(B) Fig and implementation details are provided in Table 1.

**Modifications for cell age.** The `Cell` and `TissueCell` classes were modified to add a method to set cell age. The helper `MakeTissueHelper`, which manages cell division, was updated to set daughter cell age equal to the age of the mother cell when cell division occurs.

**Table 1. Comparison of rectangular and hexagonal geometries.**

| | Rectangular | Hexagonal |
|---|---|---|
| Grid coordinate system | $(x, y, z)$ | $(u, v, w, z)$ |
| Grid center coordinates | $(0, 0, 0)$ | $(0, 0, 0, 0)$ |
| 3D simulation offset | offset: $(+x, +y)$ <br> *always alternate between offset and no offset* | offset $a$: $(-u, +w)$ <br> offset $b$: $(+u, -v)$ <br> *offset a always has offset b above* <br> *offset b always has no offset above* |
| Neighboring locations | *coordinates in brackets indicate offset of the layer*: [no offset, offset] | *coordinates in brackets indicate offset of the layer*: [no offset, offset a, and offset b] |
| *same layer* | `(+1,0,0), (-1,0,0), (0,+1,0), (0,-1,0),` | `(0,+1,-1,0), (0,-1,+1,0), (-1,+1,0,0), (+1,-1,0,0), (-1,0,+1,0), (+1,0,-1,0)` |
| *layer above* | `(0,0,+1), [(-1,0,+1), (+1,0,+1)], [(0,-1,+1), (0,+1,+1)], [(-1,-1,+1), (+1,+1,+1)]` | `(0,0,0,+1), [(+1,0,-1,+1), (-1,+1,0,+1), (0,-1,+1,+1)], [(0,+1,-1,+1), (-1,0,+1,+1), (+1,-1,0,+1)]` |
| *layer below* | `(0,0,-1), [(-1,0,-1), (+1,0,-1)], [(0,-1,-1), (0,+1,-1)], [(-1,-1,-1), (+1,+1,-1)]` | `(0,0,0,-1), [(-1,+1,0,-1), (0,-1,+1,-1), (+1,0,-1,-1)], [(0,+1,-1,-1), (-1,0,+1,-1), (+1,-1,0,-1)]` |
| Molecular lattices | Rectangular lattice where each rectangular grid location corresponds to four rectangular lattice locations, indexed clockwise from top left | Triangular lattice where each hexagonal grid location corresponds to six triangular lattice locations, indexed clockwise from upper center |

**Modifications for nutrient dynamics.**    Two additional environment components—`PulseComponent` and `CycleComponent`—were added to support pulse and cyclic nutrient profiles, respectively.

The pulse component resets the concentration of the pulse molecule (`PULSE_MOLECULE`) to the initial concentration at the specific pulsing interval $t^*$ (`PULSE_INTERVAL`). Between pulses, the component updates the available concentration of the pulse molecule $C$ based on consumption by cells and the amount of the pulse media $V$ (`MEDIA_AMOUNT`):

$$C^t = \begin{cases} C_0 & \text{if } t \bmod t^* = 0 \\ \frac{N^{t-1} - n^t}{V} & \text{otherwise} \end{cases}$$

where $C_0$ is the initial concentration of the pulse molecule, $N$ is the total number of pulse molecules, $n$ is the number of pulse molecules consumed, and $t$ is time in days.

The cycle component gradually increases or decreases the concentration of the cycle molecule (`CYCLE_MOLECULE`) over time using a modified sawtooth function with three peaks per day. The component scales the concentration of the cycle molecule $C$ as a function of time:

$$C^t = C_0 \cdot \left[ \frac{\tanh\left[(3t - \lfloor 3t \rfloor - 0.5) \cdot \sigma\right]}{2 \cdot \tanh\left(0.5 \cdot \sigma\right)} + \lfloor 3t \rfloor + 1.5 - 3t \right]$$

where $C_0$ is the initial concentration of the cycle molecule, $\sigma$ is the smoothing factor (set to 15), and $t$ is time in days.

**Model agents.**    In the model, cell agents represent healthy or cancerous tissue cells. Cells are assigned a state—quiescent, migratory, proliferative, apoptotic, necrotic, senescent, and undecided—which guides the rules the cell follows at each time point, including transitions into other states. Each cell agent contains two modules that further guide cell behavior: the metabolism module controls nutrient uptake and growth while the signaling module guides decisions between the proliferative and migratory states based on extracellular signaling. Detailed descriptions of cell states, metabolism, and signaling modules are provided in [6].

Simulations use the same healthy tissue and cancerous cell populations as introduced in [7]. The cancerous cells have crowding tolerance (`MAX_HEIGHT`) set to + 50% of baseline, metabolic preference (`META_PREF`) set to + 50% of baseline, and migratory threshold (`MIGRA_THRESHOLD`) set to −50% of baseline. Healthy tissue cell parameters were unmodified. No other cell implementation details or parameters were modified for this study.

**Model environment.**    The model environment simulates diffusion of glucose, oxygen, and a signaling molecule TGF$\alpha$. Diffusion is calculated using a simplified reaction-diffusion equation:

$$\frac{\partial C}{\partial t} = \mathscr{D}\nabla^2 C$$

where $C$ is the concentration and $\mathscr{D}$ is diffusivity of the molecule in the environment.

The finite difference approximations are given by:

$$\textbf{rectangular} \quad C^{t+\Delta t} = C^t + \frac{\mathscr{D}\Delta t}{\Delta s^2}\left(\sum_{i=1}^{4} C_i^t - 4C^t\right) + \delta\frac{2\mathscr{D}\Delta t}{\Delta z^2}\left(\sum_{j=1}^{2} C_j^t - 2C^t\right)$$

$$\textbf{hexagonal} \quad C^{t+\Delta t} = C^t + \frac{4\mathscr{D}\Delta t}{3\Delta s^2}\left(\sum_{i=1}^{3} C_i^t - 3C^t\right) + \delta\frac{2\mathscr{D}\Delta t}{\Delta z^2}\left(\sum_{j=1}^{2} C_j^t - 2C^t\right)$$

where $\Delta t$ is the time step (1 second), $\Delta s$ is the distance between two adjacent lattice locations

(15 $\mu$m for both rectangular and hexagonal geometries), $\Delta z$ is the distance between layers (8.7 $\mu$m), $i$ indexes across the three triangular or four rectangular neighbors in a layer, $j$ indexes across the two neighbors above and below the layer, and $\delta$ is 0 if $H = 1$ and 1 otherwise.

To check stability of the finite difference approximation, we perform a von Neumann stability analysis:

$$\lambda = 4\mathscr{D}\Delta t\left(\frac{1}{\Delta s^2} + \delta\frac{1}{\Delta z^2}\right)$$

For stability, $0 \leq \lambda < 1$. If not satisfied, we use a pseudo-steady state approximation:

$$\textbf{rectangular} \quad C^{t+\Delta t} = \frac{1}{4 + \delta\frac{4\Delta s^2}{\Delta z^2}}\left[\sum_{i=1}^{4}C_i^t + \delta\frac{2\Delta s^2}{\Delta z^2}\sum_{j=1}^{2}C_j^t\right]$$

$$\textbf{hexagonal} \quad C^{t+\Delta t} = \frac{1}{3 + \delta\frac{3\Delta s^2}{\Delta z^2}}\left[\sum_{i=1}^{3}C_i^t + \delta\frac{3\Delta s^2}{2\Delta z^2}\sum_{j=1}^{2}C_j^t\right]$$

Molecules are introduced into the environment by a generator component:

$$C_i^{t+\Delta t} = C_i^t + G_i^t$$

where $\Delta t$ is the time step (1 minute), $i$ is the lattice coordinate, $C$ is the array containing concentrations of a given molecule, and $G$ is the array containing the amount of that molecule added by the sites, which varies depending on the specific site component used. For this study, a constant source was used. For nutrient dynamics simulations, the pulse and cyclic components modify $G$ as a function of time.

## Quantification and statistical analysis

**Emergent behavior metrics.**    Emergent behavior was quantified using three metrics: growth rate, symmetry, and average cell cycle length.

**Growth rate.**    Growth rate captures the change in diameter of the tumor colony over time. Colony diameter at each time index is calculated as the maximum diameter $D$ across all axes: $D = \max(D_x, D_y)$ for rectangular geometry and $D = \max(D_u, D_v, D_w)$ for hexagonal geometry. For 3D simulations, diameter is calculated as the maximum across all z slices. For 3DC, diameter is calculated using the $z = 0$ (center) slice.

Growth rate at time index $i$ is then determined using a least square linear fit (Python, function `polyfit` from package `numpy` with degree of 1) on the diameters in the range $[D_{i^*}, D_{i^*+1}, \ldots, D_i]$ where $i^*$ is the time index corresponding to $t = 2$ days and $i \geq i^* + 2$.

**Symmetry.**    Symmetry captures the shape of the tumor colony at each time point, ranging from 0 (not symmetric) to 1 (perfectly symmetric). For rectangular coordinates, a colony is perfectly symmetric if for each location (x, y), the corresponding three locations (-y, x), (-x, -y), and (y, -x) are all occupied. Additionally, for rectangular coordinates where (*i*) x is zero, (*ii*) y is zero, or (*iii*) x = y, four additional locations are also checked: (x, -y), (y, x), (-x, y), and (-y, -x). For hexagonal coordinates, a colony is perfectly symmetric if for each location (u, v, w), the corresponding five locations (-w, -u, -v), (v, w, u), (-u, -v, -w), (w, u, v), and (-v, -w, -u) are all occupied.

Symmetry is then calculated as:

$$1 - \frac{1}{N}\sum_{i}^{N}\frac{n_i}{\lambda}$$

where $N$ is the number of unique occupied locations, $n_i$ is the number of unoccupied corresponding locations for a unique location $i$, and $\lambda$ is the total number of possible corresponding locations (3 or 7 for rectangular, 5 for hexagonal).

For 3D simulations, symmetry is calculated separately for each z slice and averaged. For 3DC, symmetry is calculated using the $z = 0$ (center) slice.

**Cycle length.**   Cycle length captures the average cell cycle length for cells at a given time point. Cell agents track the duration between entering a proliferative state and successfully dividing to create a daughter cell agent. Cycle lengths are first averaged for each cell agent (for cells that have divided more than once), then for all agents in the simulation (excluding healthy tissue cell agents).

For 3D simulations, cell agents from all z slices are included in the calculation; for 3DC, only cells in the $z = 0$ (center) slice are included in the calculation.

**Simulation data analysis.**   Differences in emergent behavior metrics between modeling choices were assessed using two-way analysis of variance (ANOVA). We first run the two-way ANOVA with interaction with $\alpha = 0.05$.

If the interaction is significant, we then perform Simple Main Effects testing on each factor at each level of the other factor using $\alpha = 0.05$ with a Bonferroni correction. For significant effects with more than two levels, we use pairwise Tukey tests identify which pairs of level are significant at constant levels of the other factor ($\alpha = 0.01$).

If the interaction is not significant, we then repeat the two-way ANOVA without the interaction term with $\alpha = 0.05$. For significant effects with more than two levels, we use pairwise Tukey tests identify which pairs of level are significant using the full data set ($\alpha = 0.01$).

Cell and effect means for all simulations are provided in S3, S5 and S7 Tables. ANOVA and Tukey test results are summarized in S4, S6 and S8 Tables.

## Supporting information

**S1 Fig. System representation simulations.**
(PDF)

**S2 Fig. System representation emergent behavior.**
(PDF)

**S3 Fig. Cell variability simulations.**
(PDF)

**S4 Fig. Cell variability emergent behavior.**
(PDF)

**S5 Fig. Nutrient dynamics simulations.**
(PDF)

**S6 Fig. Nutrient dynamics emergent behavior.**
(PDF)

**S7 Fig. Model framework and geometry.**
(PDF)

**S1 Table. Input options used to run simulations.**
(PDF)

**S2 Table. CPU time for simulations.**
(PDF)

**S3 Table. Cell and effect means for system representation emergent metrics.** (PDF)

**S4 Table. ANOVA for system representation emergent metrics.** (PDF)

**S5 Table. Cell and effect means for cell variability emergent metrics.** (PDF)

**S6 Table. ANOVA for cell variability emergent metrics.** (PDF)

**S7 Table. Cell and effect means for nutrient dynamics emergent metrics.** (PDF)

**S8 Table. ANOVA for nutrient dynamics emergent metrics.** (PDF)

## Author Contributions

**Conceptualization:** Jessica S. Yu, Neda Bagheri.

**Formal analysis:** Jessica S. Yu.

**Software:** Jessica S. Yu.

**Supervision:** Neda Bagheri.

**Visualization:** Jessica S. Yu.

**Writing – original draft:** Jessica S. Yu, Neda Bagheri.

**Writing – review & editing:** Jessica S. Yu, Neda Bagheri.

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
