## [Decision Letter · Decision Letter 0]

26 Sep 2023

Dear Prof. Bagheri,

Thank you very much for submitting your manuscript "How modeling decisions impact biological insight: navigating the broad landscape of characterizing biology with spatial-temporal models" for consideration at PLOS Computational Biology.

As with all papers reviewed by the journal, your manuscript was reviewed by members of the editorial board and by several independent reviewers. The reviews (see below) appreciated the topic and your work, but also provide several substantive critiques that would need to be addressed before the manuscript could be considered further. Therefore, we invite the resubmission of a significantly-revised version that takes into account the reviewers' comments. 

We cannot make any decision about publication until we have seen the revised manuscript and your response to the reviewers' comments. The invitation to revise is not a guarantee of publication and we would expect the revisions/response to be comprehensive. Your revised manuscript is likely to be sent to reviewers for further evaluation.

Sincerely,

Feilim Mac Gabhann, Ph.D.

Editor-in-Chief

PLOS Computational Biology

Reviewer's Responses to Questions

**Comments to the Authors:**

Reviewer #1: This manuscript reports the outcome of several computational experiments to test the effect of certain modeling choices on model behavior. The biological setting is tumor growth, modeling in an agent-based model framework. Features tested include choices of representation of tumor geometry, representation of cells, and spatial dimension. The results reported are that there are quantitative differences in metrics such as tumor growth or cell cycle length and, in some cases, qualitative differences.

When a modeler sets out to build a model of a biological system in order to understand it better and discover new features, then certain choices need to be made. They are determined by a number of considerations, such as whether quantitative or qualitative answers to questions are required, computational resources, degree of conformity to “reality,” and others. A model can of course be wildly unrealistic and still provide valuable insights. The study reported here is intended to shed light on the effects of some choices to be made in the specific setting to be considered.

The reported results are qualitative, comparing one the outcome of one choice to another. It would be helpful in this case to compare both outcomes to “reality” in order to make an informed choice. I do not find the reported results very useful, since it is not clear how they generalize to other biological settings. It is certainly not surprising that different choices of geometry, for instance, lead to differences in model behavior, but does it make a difference if I wanted to model a kidney disease rather than cancer? As another example, knowing that 2D and 3D simulations tend to lead to different outcomes is useful, but knowing how to scale certain variables from a 2D simulation to a 3D one would be significantly more helpful.

In summary, I commend the authors for carrying out these studies. But any modeler should do the same before making these choices in order to build models that are credible for the intended purpose. However, the results reported are of limited use, as modeling platforms vary and different biological systems have different behavior leading to potentially quite different effects of model choices. For these reasons I cannot recommend publication of this manuscript.

Reviewer #2: How modeling decisions impact biological insight: navigating the broad landscape of characterizing biology with spatial-temporal models (Yu & Bagheri, 2023)

Authors introduce an agent-based modeling approach which they use to study the effect of geometry, dimension, and context (colony/tissue) on some emergent behaviors (growth rate, symmetry, and cell cycle length). The results are explained within a broader discussion on modeling decisions. The results are presented in a coherent and systematic fashion, and the figures are very well designed. This is an important paper that investigates common agent-based decisions that every modeller must make.

Minor concerns:

My main (minor) criticism of the paper is that the Introduction is rather short, and I believe some of the description later in the paper would be better suited for the Introduction section. For example, the background info in the first three paragraph of the results may be better suited within the introduction. There reference to citation #12 on page 3 is a helpful guidance to the theme of the paper and authors should also consider moving this to the introduction.

There are several vague sentences in the results, describing that a change exists, but not the direction of the change (e.g. input parameter A causes emergent behavior B to rise/fall). Please clarify the following sentences w/ more concrete result:

1. Page 1, first paragraph: “...such models and results are not readily generalizable to other cell types or contexts.” What context? Environmental context? Why is not generalizable? Some further description would be helpful.

2. Page 3, second paragraph: “Qualitative trends in emergent behavior across different choices of geometry and dimension are largely consistent over time” – what trends, and what exactly is consistent?

3. Page 8, second paragraph: “Tumor shape is strongly impacted by glucose level” – how does shape change? Or is it simply size?

Figure 5: there appears to be little effect of nutrient level (low/basal/high) in figure 5B. Why is that? This is not noted within the text.

There is an extensive literature on ABM’s in math biology, which generally seems under-cited in the paper. Consider citing some of the following ABM reviews (or references contained therein):

1. Hybrid modeling frameworks of tumor development and treatment (PMID: 31313504)

2. Agent-based methods facilitate integrative science in cancer (PMID: 36404257)

3. Uncertainty and sensitivity analyses methods for agent-based mathematical models: An introductory review (Hamis et al)

4. A review of cell-based computational modeling in cancer biology (PMID: 30715927)

Reviewer #3: This paper predominantly centers around the utilization of an ABM named ARCADE. However, there are several major concerns that have tempered my enthusiasm for the paper. These concerns are outlined below:

The title and abstract are misleading. Initially, I believed I was reading a review of modeling strategies rather than an exploration of a single modeling approach in a specific scenario. The title and abstract should be rephrased to accurately reflect the content presented in the paper.

The authors place significant emphasis on "emergent behavior." However, this term remains undefined, and none of the results seem to exhibit emergent behavior in the classical physical sense. Therefore, more precise and accurate terminology should be employed to describe the observed phenomena.

Numerous ABM platforms have been employed and published in various fields, ranging from cancer research to stem cell studies and beyond. It is unclear from this reading how the work presented relates to these other studies. The paper should clarify its contributions and distinctions in relation to existing research.

**Have the authors made all data and (if applicable) computational code underlying the findings in their manuscript fully available?**

Reviewer #1: Yes

Reviewer #2: None

Reviewer #3: Yes

PLOS authors have the option to publish the peer review history of their article (what does this mean?). If published, this will include your full peer review and any attached files.

Reviewer #1: No

Reviewer #2: No

Reviewer #3: No
---

## [Decision Letter · Decision Letter 1]

12 Feb 2024

Dear Prof. Bagheri,

We are pleased to inform you that your manuscript 'Model design choices impact biological insight: Unpacking the broad landscape of spatial-temporal model development decisions' has been provisionally accepted for publication in PLOS Computational Biology.

Best regards,

Feilim Mac Gabhann, Ph.D.

Editor-in-Chief

PLOS Computational Biology

Reviewer's Responses to Questions

**Comments to the Authors:**

Reviewer #2: Thank you for addressing all previous concerns

**Have the authors made all data and (if applicable) computational code underlying the findings in their manuscript fully available?**

Reviewer #2: Yes

PLOS authors have the option to publish the peer review history of their article (what does this mean?). If published, this will include your full peer review and any attached files.

Reviewer #2: No

---

## [Editor Report · Acceptance letter]

5 Mar 2024

PCOMPBIOL-D-23-00491R1 

Model design choices impact biological insight: Unpacking the broad landscape of spatial-temporal model development decisions

Dear Dr Bagheri,

I am pleased to inform you that your manuscript has been formally accepted for publication in PLOS Computational Biology. Your manuscript is now with our production department and you will be notified of the publication date in due course.

With kind regards,

Zsofia Freund
